# Modeling Glioblastoma with Brain Organoids: New Frontiers in Oncology and Space Research

**DOI:** 10.3390/ijms262110664

**Published:** 2025-11-01

**Authors:** Laura Begani, Luigi Gianmaria Remore, Stefania Ragosta, Massimiliano Domenico Rizzaro, Laura Guarnaccia, Giovanni Andrea Alotta, Laura Riboni, Monica Rosa Miozzo, Emanuela Barilla, Chiara Gaudino, Marco Locatelli, Emanuele Garzia, Giovanni Marfia, Stefania Elena Navone

**Affiliations:** 1Center for Aerospace Medicine and Advanced Therapy (CeMATA), Neurosurgery Unit, Fondazione IRCCS Ca’ Granda Ospedale Maggiore Policlinico, 20122 Milan, Italy; laura.begani@policlinico.mi.it (L.B.); stefania.ragosta@policlinico.mi.it (S.R.); laura.guarnaccia@policlinico.mi.it (L.G.); stefania.navone@policlinico.mi.it (S.E.N.); 2Neurosurgery Unit, Fondazione IRCCS Ca’ Granda Ospedale Maggiore Policlinico, 20122 Milan, Italy; luigi.remore@unimi.it (L.G.R.); massimiliano.rizzaro@unimi.it (M.D.R.); marco.locatelli@policlinico.mi.it (M.L.); 3Department of Pathophysiology and Transplantation, University of Milan, 20122 Milan, Italy; 4Andremacon Srl, Viale Ortles 22/4, 20139 Milan, Italy; g.alotta@andremacon.com (G.A.A.); l.riboni@andremacon.com (L.R.); e.barilla@andremacon.com (E.B.); 5Medical Genetics Unit, ASST Santi Paolo e Carlo, 20142 Milan, Italy; monica.miozzo@unimi.it; 6Department of Health Sciences, University of Milan, 20122 Milan, Italy; 7Department of Neuroradiology, Azienda Ospedaliero-Universitaria Policlinico Umberto I, 00161 Rome, Italy; c.gaudino@policlincoumberto1.it; 8Aerospace Medical Institute “A. Mosso”, 20138 Milan, Italy; emanuele.garzia@asst-santipaolocarlo.it; 9Reproductive Medicine Unit, Department of Mother and Child, San Paolo Hospital Medical School, ASST Santi Paolo e Carlo, 20142 Milan, Italy

**Keywords:** glioblastoma, preclinical models, organoids, space, microgravity

## Abstract

Glioblastoma (GBM) is the most malignant primary brain tumor, characterized by extensive heterogeneity, invasiveness, infiltrating behavior, and resistance to standard therapies, including radiation and temozolomide (TMZ). Despite considerable efforts in investigating its pathophysiology, GBM represents one of the most challenging cancers to treat, with a median survival rate under 15 months and a 5-year survival rate below 5%. A major barrier to progress in GBM therapy development is the lack of reliable preclinical models that faithfully recapitulate the tumor’s molecular heterogeneity, invasive behavior, and complex microenvironment. Traditional cell lines and xenograft models often fail to reflect the key pathological features of human GBM, including immune suppression, vascular abnormalities, and treatment resistance. In recent years, attention has focused on the development of numerous clinically relevant GBM models based on brain organoids as a powerful “disease-in-a-dish” model. They strongly mimic GBM key histopathological and molecular features, such as the tumor’s cellular heterogeneity, genetic landscape, and microenvironment, enabling more accurate studies of tumor biology, invasion, and therapeutic response in a controlled in vitro setting. Notably, research in microgravity offers a unique and promising platform to study cancer biology under conditions that enhance tissue self-organization, mimic aspects of tumor growth, and potentially unveil novel therapeutic vulnerabilities. This review compares organoids to conventional preclinical models, tracing their historical development and salient features, focusing on the preparation and use of organoids in GBM research. We also introduce a novel and promising field of organoid application: space-based organoid brain research.

## 1. Background

Glioblastoma (GBM) is the most aggressive and frequent primary brain tumor in adults, and recent research has proposed that neural stem cells and glial precursor cells found in the subventricular zone may be its true precursors, rather than mature astrocytes [1].

According to the World Health Organization (WHO) guidelines, it is classified as a grade 4 astrocytoma with wild-type isocitrate dehydrogenase 1 or 2 (*IDH1/2*) alleles often in association with other genetic alterations, including amplification of chromosome 7 and deletion of chromosome 10, *EGFR* amplification, and *TP53* mutation [2].

Relapses 6 to 12 months after the treatment are very common, and the prognosis for survival is usually poor [3].

GBM is characterized by a marked intertumoral and intratumoral heterogeneity at the cellular, epigenetic, and genetic levels, rapid progression, invasivity, abnormal angiogenesis, and resistance to radio- and chemotherapy [4]. In particular, therapy resistance is due to a subpopulation of tumorigenic stem-like cells, known as GBM stem cells (GSCs), which are self-renewing, pluripotent, highly proliferative, and genetically unstable [5]. GSCs drive tumor progression and, upon pro-tumorigenic stimuli, can transdifferentiate into endothelial cells, directly contributing to the formation of abnormal blood vessels and GBM growth [6,7].

Because of its molecular and genetic features, the complex microenvironment, as well as the absence of an effective therapy, GBM represents a considerable therapeutic challenge for which novel paradigms and models should be considered.

A key obstacle significantly impeding progress in GBM treatment is the absence of appropriate and reliable human in vitro models that adequately mirror the heterogeneity and complexity of GBM compared to current 2D models and faithfully mimic human physiological and pathological processes, overcoming the problem of species and the different genetic backgrounds of mouse models [8]. This gap in GBM translational modeling significantly hampers the predictive value of preclinical studies and contributes to the high failure rate of candidate therapies in clinical trials. However, organoids are emerging as a promising tool in the context of GBM research, since they can reproduce the brain microenvironment while maintaining tumor heterogeneity. Indeed, organoids can be established under highly defined and reproducible conditions, allowing researchers to systematically modulate matrix properties, cellular composition, and signaling factor gradients. This degree of control enables the dissection of tumor biology with unprecedented precision, fostering insights into mechanisms of invasion, resistance, and plasticity that are often masked in more complex organisms. Furthermore, these models provide a powerful platform for high throughput and rapid drug testing, accelerating the translational pipeline by enabling the early-stage evaluation of therapeutic efficacy and toxicity in a physiologically relevant context. Therefore, GBM organoids hold the promise to bridge the gap between 2D cell cultures and in vivo models [9].

In this review, we summarize the development of GBM preclinical models, from traditional 2D culture to advanced 3D models, focusing on organoids, their unique characteristics and applications, and the main methodologies for generating them, highlighting their impact on GBM research and knowledge. Moreover, we discuss the effects of space-based factors on organoids during space missions and the importance of these models. Organoids not only offer a platform to unravel fundamental biological processes underlying human health and disease, but open new perspectives in oncology. The altered microenvironment of space can accelerate the discovery of tumor vulnerabilities and drug responses not readily captured on the Earth. By leveraging insights gained from space-based experimentation, we aim to highlight the translational potential of space-grown organoids as powerful, next-generation tools for improving cancer research, developing innovative therapeutic strategies, and improving clinical outcomes.

## 2. Conventional Approaches in GBM Research

### 2.1. 2D Models

In GBM research, two-dimensional (2D) models are the most widely used experimental systems due to their cost-effectiveness and ease of manipulation. Two-dimensional cultures are suitable for preliminary experiments since they are compatible with a broad range of standard laboratory techniques, as well as for the high-throughput screening of potential drugs.

They have been developed to better comprehend the distinctive features of GBM and to display the advantages and disadvantages that must be considered in experimental settings [10].

The cell lines typically used in GBM research are C6 [11], U87MG, and U251MG [12], grown in a serum-supplemented medium on flat plastic surfaces (Figure 1A). They were developed several years ago by exposing animals to chemicals and oncogenic substances or by artificially manipulating patient-derived cells to induce unlimited proliferation [13].

The C6 cell line is considered one of the firs models in neuro-oncology research. First introduced in 1960, it was created by repetitively exposing Wistar rats to N-methyl nitrosourea. These cells show features similar to human glioma cells, including the increased expression of PDGF-β, EGFR, IGF-1, and a boosted Ras activity. However, *IDH1* and *IDH2* mutations are not present [14,15].

The human-derived cell lines U87MG and U251MG were introduced in the late 1960s. Both resulted as a response to TMZ [16,17] and adequately recapitulate the GBM genetic profile, including mutant PTEN, upregulation of PI3K and Akt, and alteration in cell cycle control [18].

Overall, they represent simple and reproducible systems, allowing for studies on specific tumor markers, prognosis-related factors, and molecular pathways involved in GBM pathogenesis [10]. They are easy to maintain, provide fast preliminary data about proliferation, invasion, and migration, and their use is not subject to ethical concerns.

However, a number of shortcomings must be considered: since they are constituted only by tumor cells, they do not recapitulate the GBM microenvironment and interactions with non-malignant cells. Moreover, they fail in mimicking heterogeneity and spatial organizations, as well as nutrient concentrations and the gradient of oxygen [10,19].

### 2.2. In Vivo Models

Traditional in vivo systems include genetically engineered mice (GEM) (Figure 1B) and xenografts models (Figure 1C,D).

Since GBM pathogenesis and prognosis are strictly related to a number of well-established mutations, transgenic/knockout mice were developed to investigate the effects concerning the activation of oncogenic signaling pathways (EGFR, RAS, PI3K) and the loss of GBM oncosuppressors (TP53, PTEN) [20,21]. For example, Ding et al. developed a transgenic mouse model expressing the oncogene v-Ha-ras, and they found that the level of this oncogene was directly proportional to the expansion of the tumor [22].

However, these models fail in reproducing the genetic and phenotypic heterogeneity of GBM, as well as tumor microenvironment (TME), because non-malignant normal host cells are missing [23].

Xenograft models, including cell line-derived xenografts (CDX) and patient-derived xenografts (PDX), represent useful in vivo systems to study GBM biology and development.

CDX models involve the implantation of established immortalized GBM cell lines (e.g., U87-MG, LN229) into immunocompromised mice, either subcutaneously or orthotopically, while PDX models involve the direct engraftment of patient-derived tumor tissues in immunodeficient mice [20,24].

Unlike GEM and CDX models, PDX models faithfully recapitulate the parental tumor architecture, as well as the histopathological features, including invasive behavior and therapy resistance [25].

However, the costs and the lack of a human brain microenvironment represent significant disadvantages that limit the use of these models in GBM research [26].

In order to overcome this problem, a valid alternative could be the use of canine GBM models that spontaneously grow with an incidence similar to that reported in humans [27]. In fact, it has been demonstrated that the gliomas detected in canines mirror those seen in humans for anatomy and pathophysiology [19]. However, several ethical concerns must be considered, together with the need to strictly observe the 3Rs (reduction, replacement, refinement) [19] (Figure 1E).

## 3. In Vitro 3D Models: The Bridge Between Past and Future of GBM Research

### 3.1. Spheroids

In 2006, Lee et al. boosted the transition from traditional 2D cell lines to more sophisticated preclinical models with an important study on GBM primary cells obtained from the same patient and cultured under two different conditions: in a medium with serum as cell lines or in a medium supplemented with growth factors suitable for GSCs growth. The results were remarkable. The GSCs showed remarkable genotypic and phenotypic similarities with the parental tumor and, when implanted in mice, were more invasive compared to the cells cultured with serum [28]. GSCs play a pivotal role in maintaining the growth of tumors and supporting therapy resistance due to their abilities of self-renewal and differentiation [29,30], and this study highlighted the effective possibility of using them to investigate GBM biology.

In order to preserve GSCs’ subpopulation, patient-derived primary GBM cells are generally cultivated under floating conditions, in serum-free media supplemented with growth factors, including EGF/bFGF. Under these conditions, GSCs grow as neurospheres and represent the most common type of GBM spherical cancer models, first described by Singh et al. in 2003 [31].

Spheroids are three-dimensional (3D) cell culture systems and represent the first step of GBM research in the third dimension. They are constituted by the heterogeneous aggregates of different cancer cell types not attached to any solid surface for support, allowing the growth and proliferation of floating cell clusters with a spheroid or spheroid-like architecture [32].

They can be cultivated in specific gels, for example Matrigel^®^ (Corning, NY, USA) that mimic the properties of the extracellular matrix (ECM) or, alternatively, in enriched EGF/bFGF media in non-coated plates (Figure 2A). Spheroids show significant advantages compared to 2D cultures: they can be maintained for a long time, while retaining the main features of the original tumor, including the genetic in vivo profile, cell–cell interactions, and the GBM architecture with a necrotic non-proliferating core, a middle layer with quiescent cells, and an outer proliferating layer in close contact with nutrients [33]. Moreover, it has been demonstrated that single cells dissociated from neurospheres keep the ability to generate infiltrating GBM when implanted in animals. They also accurately mirror spatial organization and tissue polarity [32,33].

However, spheroids show crucial disadvantages that could limit their use. They fail in accurately mirroring the TME and interactions between GBM cells and non-tumor cells since they are completely constituted of cancer cells. They also fail to recapitulate intratumoral heterogeneity when cultured for long periods, and cells exhibit genomic and transcriptional changes with the loss of quiescent GSCs after 20–30 passages [19,34].

### 3.2. Organoids

Spheroids progressively improved and evolved into more sophisticated 3D models, known as organoids. An organoid is a miniaturized and simplified version of an organ produced in vitro in three dimensions, which accurately mimics the key functional, structural, and biological complexity of that organ [35]. These structures, also defined as “mini brains”, represent a revolutionary tool in the field of brain disorders research as they can faithfully recapitulate in vivo human organ physiology and histology [19].

In 2013, Lancaster et al. were the first to create a human cerebral organoid from embryonic bodies derived from pluripotent stem cells [36], following principles consistent with previous findings on ECM organization and tissue structuring [37]. Developed within an ECM, such as Matrigel^®^, they exhibited characteristics of the human cerebral cortex and recapitulated features of various brain regions [38] (Figure 2B).

In the last few years, organoids have emerged as promising in vitro models for studying the key features of GBM due to their ability to recapitulate the complex three-dimensional architecture, tumor heterogeneity, and surrounding microenvironment, which significantly influence tumor growth and invasiveness [23].

Several protocols and methodologies have been developed in order to model GBM, starting with different sources, including tissue specimens, gene-edited stem cells alone or co-cultured with organoids, and patient-derived stem cells [2,23].

The main types, their characteristics, and applications are described in the following sections (Figure 3).

#### 3.2.1. GBO: Glioblastoma Organoid

In 2016, Hubert et al. [39] set up a new protocol for generating organoids from patient-derived GBM cells, based on a previous procedure for a cerebral organoid described by Lancaster et al. [36]. Surgical specimens were minced to obtain a suspension of GSCs, then embedded in Matrigel^®^ and cultured in a complete medium for about 2 months. The resulting GBOs displayed a marked heterogeneity, with rapidly dividing cells in the outer region and a hypoxic core characterized by GSCs resistant to therapy. This was the first complex model composed of different types of cells compared to homogenous spheroids [39,45]. This pioneering procedure was improved upon by Jacob et al., in 2020, with a faster protocol (2 weeks instead of 2 months): GBOs were created directly from GBM pieces, without single-cell dissociation step, exogenous neurotrophic factor, and extracellular matrix [40]. Tumor fragments were cultured in ultra-low attachment plates containing a specific GBO medium composed of 50% of DMEM F12 and Neurobasal™ medium supplemented with NEAAs, N2, B27, human insulin, and 2-mercaptoethanol. Plates were placed on an orbital shaker (speed rotation 120 rpm) in an incubator at 37 °C, 5% CO_2_, and 90% humidity. Since, in the first days of the culture, tumor pieces released cellular debris, the medium was changed every 2 days, and within 1–2 weeks they formed spherical organoids. In order to avoid core necrosis and to guarantee an optimal oxygen diffusion, GBOs cultured for more than 1 month were cut to ~200–500 µm diameter pieces [40].

Jacob et al. demonstrated that brain organoids derived from pluripotent stem cells faithfully recapitulate both the genetic heterogeneity and the complex histoarchitecture of GBM as driver gene expression (e.g., *EGFR* mutations), offering a robust platform for the development of targeted and personalized therapies. Organoids were generated from the tumor samples of 52 patients, of which 66.7% harbored the IDH1 mutation and 75% were recurrent tumors. Eight organoids were successfully engrafted into murine brains, retaining the aggressive phenotype and genetic identity of the original tumors. Mimicking post-surgical treatment, the authors tested both standard and targeted therapies, including CAR-T cell therapy. The study revealed variable therapeutic responses among organoids, correlating with patient-specific genetic alterations. Notably, CAR-T therapy targeting EGFRvIII showed pronounced efficacy in organoids derived from patients carrying the EGFRvIII mutation [40].

A crucial characteristic of GBM is the presence of a tumor microenvironment made up of various stromal components, including tumor-associated macrophages, dendritic cells, endothelial cells, neurons, and reactive astrocytes necessary for invasion, cytotoxic resistance, and tumor growth [46]. While in GBOs previously developed by Hubert et al., no immune cells were found, Jacob et al. demonstrated the presence of a variety of non-neoplastic cell types, including immune cells, stromal cells, oligodendrocytes, endothelial cells, and neurons, after two weeks in culture.

Interestingly, in 2024 Liang and He collected the most recent papers on the main classes of GBM organoids published in four databases (Wiley Online Laboratory, PubMed, Embase, and Web of Science) with the aim of clarifying the most commonly used model in GBM research: the review, conducted on 42 articles, showed that the direct cultivation of tumor tissue specimens was the most frequently used method, followed by co-culturing and gene-editing, highlighting the broader applications and strengths of GBOs rather than other methods [47].

#### 3.2.2. NeoCOR: Neoplastic Cerebral Organoid

In the context of brain organoid-based models in GBM research, stem cells play a pivotal role in recapitulating the tumor and the organ where it develops, showing a faithful representation of the brain with structures composed of different cells similar to those found during brain development. Several protocols have been established to generate cerebral organoids (COs) following the specific steps described by Lancaster et al. Human PSCs were trypsinized into single cells and cultured in a 96-well U-bottom plate, in a medium supplemented with bFGF, in order to generate embryoid bodies (EBs), three-dimensional aggregates subjected to neural induction in a specific medium that promote the development of neuroectoderm instead of mesoderm or endoderm. Neuroectodermal tissues were then embedded in Matrigel^®^ droplets to induce self-organization and to mimic a three-dimensional structure, in order to allow the cerebral organoids to mature under agitation in a spinning bioreactor or orbital shaker [36].

The first stem cell-derived GBM organoid was generated from PSCs genetically engineered at different stages of CO formation in order to introduce the driver genetic mutations found in GBM, involving genes such as *TP53*, *EGFR*, *TERT*, *PTEN*, and *CDKN2A/B*.

The first model was set up by Ogawa et al., in 2018 [41], through the genome-editing technique CRISPR/Cas9, to investigate tumor initiation and progression. In particular, COs were developed from the human embryonic stem cell (ESC) line H9 according to the above-mentioned protocol. Organoids were cultured for a period of four months to allow their maturation, and then genetically modified by introducing the HRasG12V oncogene precisely into the *TP53* locus via electroporation [41]. Since the transgene was co-expressed by the fluorescent marker tdTomato, the authors were able to monitor tumor behavior and, notably, they found that the mutated cells acted as GBM cells, proliferating and invading healthy organoids. Furthermore, when implanted in immunodeficient mice, they showed aggressive behavior [41,48].

This model was further studied by Bian et al., who introduced a new term for these organoids: neoplastic cerebral organoids (neoCORs) [42]. Here, different clinically relevant oncogenic mutations were introduced via electroporation by CRISPR/Cas9 technology at the end of the neural induction stage, before the Matrigel^®^ embedding. EBs were nucleofected with plasmids, then included in droplets of Matrigel^®^ and cultured on an orbital shaker to allow for organoid maturation. The resulting models were made up of normal and pathological tissues, unlike GBOs, allowing studies on GBM invasiveness as well as interactions between tumor cells and brain microenvironment [42]. Moreover, by introducing specific mutations into organoids generated with patient-derived cells, neoCORs can also be employed to assess patient-specific susceptibility to different combinations of driver mutations.

NeoCORs represent a powerful tool for studies on GBM biology and pathophysiology, as well as a promising model for targeted drug testing. However, some weaknesses must be considered: they are slow-growth models, and CRISPR/Cas9 use is restricted to oncogenes and mutations that have been studied to date.

#### 3.2.3. GLICO: Glioblastoma Cerebral Organoid

GLICO models can be considered to be a hybrid model, combining the features of GBOs and neoCORs.

The first GLICO model was developed by da Silva et al. by co-culturing human GBM spheroids (grown from GSCs) with mouse ESCs [43]. Then, it was improved by Linkous et al., who developed a new GLICO model by culturing GFP-labelled patient-derived GSCs with human COs [44]. COs were generated following the Lancaster protocol described previously. GSCs obtained from tumor dissociation were transfected with GFP and cultured in an NBE medium (Neurobasal™ medium with B27, N2, EGF, and bFGF). Individual COs with at least 1 month of maturation were placed in a 24-well plate, one organoid per well, and GSCs expressing GFP were distributed in each organoid-containing well. Plates were incubated at 37 °C for 24 h without agitation [44]. Since patient GSCs are required, they accurately recapitulate interpatient variability, a fundamental feature in the context of personalized medicine: in fact, Linkous et al. found that each GLICO model generated from different patient-GSCs showed different sensibility to chemotherapy and radiation. In addition, GLICO models faithfully mimic the invasive and proliferation pattern of the parental tumor, also providing information about tumor–brain interactions [44].

Interestingly, Fedorova et al. developed a GLICO model to investigate GBM cell migration and invasive behavior by co-culturing mature COs, U87MG-GFP, and tdTomato. They demonstrated that the presence of ECM components, such as Matrigel^®^ or Geltrex™, significantly enhanced GBM cell migration compared to simple GLICO without ECM proteins, and introduced a robust workflow for the visualization and quantification of GBM cell infiltration within cerebral organoids, employing confocal microscopy combined with tissue clearing techniques, which can be adapted for the detailed analysis of tumor invasion dynamics [49].

However, as mentioned for previous models, the main disadvantage of these models is in the lack of vascularization and immune cells that play a pivotal role in influencing GBM growth and aggressiveness in vivo [9,43,44].

Overall, organoid research experienced rapid methodological and conceptual growth between 2016 and 2020 (Figure 3). This period was marked by remarkable technological advances that enhanced organoid fidelity, scalability, and clinical relevance across multiple biological systems. In 2016–2017, the introduction of patient-derived organoids revolutionized cancer modeling by enabling the preservation of individual tumor heterogeneity and personalized therapeutic assessment. A notable milestone during this period was the development of glioblastoma organoids (GBOs), which provided a physiologically relevant model for studying one of the most aggressive and treatment-resistant brain tumors. Foundational studies demonstrated that GBOs retain the key histopathological and molecular characteristics of primary glioblastoma, including intratumoral heterogeneity, self-renewing glioma stem-like cell populations, and invasive growth patterns resembling those observed in vivo.

By 2018, combining organoid platforms with CRISPR/Cas9 gene editing, the field expanded from purely patient-derived organoids to genetically engineered organoid models (from healthy cerebral organoids) to model tumor initiation and specific driver combinations. The above-mentioned categories of complementary GBM models expanded the translational potential for disease modeling, high-throughput drug screening, and personalized medicine.

Building on this progress, between 2018 and 2019, the GLICO model, which co-cultures patient-derived glioblastoma stem cells with human cerebral organoids, was introduced, enabling the examination of tumor–brain interactions, invasion dynamics, and neural network disruptions in a three-dimensional context.

Finally, in 2020, a significant milestone was reached with the introduction of GBM models generated directly from surgical specimens, thereby allowing the introduction of a rapid and reproducible protocol as well as the creation of a large biobank that supported personalized therapy testing.

Collectively, these developments transitioned GBM organoid technology from proof-of-concept models to translationally relevant platforms, bridging the gap between conventional cell culture and patient-derived xenografts.

The characteristics of the different GBM models discussed above are summarized in Table 1.

## 4. Clinical Applications of GBM Organoids

Brain tumor organoids, particularly those modeling GBM, represent a transformative and patient-specific platform for precision neuro-oncology, faithfully preserving the histological, genetic, and cellular heterogeneity inherent in this aggressive malignancy. As previously explained, traditional two-dimensional cell cultures and animal models frequently fail to replicate the invasive growth patterns and treatment resistance characteristic of GBM. By contrast, GBM organoids sustain three-dimensional architecture and the difficulty of drug penetration, offering a more physiologically relevant substrate for therapeutic investigation. Recent efforts to integrate these organoids within microfluidic organoid-on-a-chip systems have further enhanced their translational potential. In particular, microfluidic platforms have enabled the vascularization of GBM models and simulation of the blood–brain barrier (BBB), facilitating studies of therapeutic delivery and tumor–vasculature interactions with a degree of realism unachievable in static models. One exemplary model comprises a vascularized GBM spheroid embedded within a microfluidic device, co-cultured with iPSC-derived endothelial cells, pericytes, and astrocytes to generate perfusable vasculature. This BBB-GBM platform allowed tumor spheroids derived from GBM patient-derived xenograft lines to grow in intimate contact with vascular networks that mimic in vivo co-option, demonstrating preserved vessel integrity and tight-junction expression even in proximity to tumor masses. Such microfluidic systems have been effective in quantifying parameters like vascular permeability and receptor expression (e.g., LRP1) that regulate nanoparticle-mediated drug transport across the BBB [50]. Additionally, organotypic microfluidic models recapitulating GSCs vascular niche have been developed. These systems employ hydrogel-based scaffolds and perfusable microvascular networks to study the perivascular microenvironment that sustains GSCs stemness, invasion, and therapeutic resistance. Notably, Truong et al. identified *CXCL12–CXCR4* signaling as a driver of GSCs migration and invasive morphology within these devices. Moreover, they used AMD310, a pharmacological antagonist of *CXCL12–CXCR4* signaling, to effectively inhibit invasion, demonstrating the capacity for mechanistic insight and drug screening in a physiologically relevant GBM microenvironment. Such molecules might be applied as a screening tool on patients’ serum in the future [51]. Extending these models, emerging biosensor-enhanced organoid-on-a-chip platforms enable real-time monitoring of GBM tumor microenvironment dynamics, including cell migration, proliferation, and intercellular interactions, providing unprecedented resolution and temporal control during drug testing [52]. Furthermore, tumor organoid-on-a-chip systems, which unify patient-derived organoids with microfluidic engineering, allow for the precise modulation of nutrient gradients, mechanical stimuli, and perfusion, thereby enhancing physiological relevance and predictive accuracy in therapeutic screening. These hybrid platforms support high-throughput drug testing, immunotherapeutic evaluation, and metastasis modeling, although standardization and scalability remain key challenges to be addressed.

For instance, Shi et al. developed an in vitro microfluidic “BBB-U251” chip that integrates primary human brain microvascular endothelial cells, pericytes, astrocytes, and U251 glioma cells. The BBB compartment on the chip showed selective permeability to FITC-dextran of different molecular weights and to three model drugs, demonstrating functional barrier behavior. Then, six active components from traditional Chinese medicine were introduced via the “blood channel”, their permeation quantified by HPLC-UV, and their antineoplastic efficacy was measured on the 3D glioma culture. This study revealed that permeability coefficients more closely resembled in vivo values, and that the antineoplastic effects were significantly reduced in the chip compared to the direct glioma culture due to the barrier. The authors, therefore, concluded that the model better mimics in vivo delivery challenges and is a promising platform for brain tumor drug screening [53].

On the other hand, three-dimensional bioprinting is rapidly transforming GBM-on-a-chip technology, enabling the construction of personalized, physiologically relevant tumor microenvironments that facilitate high-fidelity drug testing. Recent methodologies have implemented extrusion-based bioprinting to fabricate concentric ring architectures incorporating patient-derived GBM cells, endothelial cells, and brain-derived ECM to recapitulate the structural, biochemical, and mechanical heterogeneity of GBM, including the establishment of radial oxygen gradients akin to in vivo tumor hypoxia [54]. In this context, Yi et al. pioneered a vascularized GBM model by using a multi-cell bioink comprising patient-derived GBM cells, astrocytes, microglia, and vascular components, integrated into a perfusion chip via hollow channels [55]. This platform enabled sustained flow, live imaging, and assessment of drug responses, revealing differential sensitivity to temozolomide compared to conventional 2D models, likely due to microenvironmental context. Beyond structural mimicry, 3D bioprinted GBM microenvironments have served as robust platforms to interrogate drug resistance. For example, Tang et al. fabricated tetra-culture constructs combining GSCs with macrophages within bioprinted hydrogels and showed that they exhibited enhanced resistance to EGFR inhibitors and temozolomide, compared to simpler cultures [56]. Drug penetration dynamics were validated using fluorescent dextran diffusion assays, and predictive transcriptomic analyses identified candidate compounds, such as abiraterone and ifosfamide, that effectively reduced tumor growth in the xenograft models. This model, therefore, demonstrated a mechanism of tumor resistance mediated by the microenvironment.

Going beyond three-dimensional modeling, 4D bioprinted arrays—employing thermo-responsive shape-memory polymers that transition from 3D cell culture inserts into histological cassettes—have been engineered for rapid, functional drug evaluation in GBM patient-derived organoids. These arrays support high-throughput assessment of drug sensitivity, on-target activity, and combination synergy, thus streamlining preclinical drug evaluation pipelines. For instance, Chadwick et al. introduced a novel 4D printed, programmable cell culture array using a thermo-responsive shape-memory polymer (SMP) for GBM drug testing. Fabricated via high-precision projection micro-stereolithography (PμSL), the SMP constructs self-transform between 3D cell culture formats and histology-compatible cassettes, enabling streamlined, high-throughput analysis without manual sample transfer. The platform supported the culture of patient-derived GBM organoids (PDOs), preserving intratumoral heterogeneity and tumor microenvironment elements, while enabling testing of both standard and targeted therapies. Notably, the authors identified synergistic interactions between niraparib and BEZ235, while also observing BEZ235-mediated protection against TMZ toxicity, underscoring the utility of the platform in identifying clinically actionable therapeutic combinations [57]. Finally, advances in biofabrication have led to 96-well bioprinted neurovascular unit (NVU) models incorporating patient-derived GBM spheroids, vascular endothelial cells, and functional microvascular networks. These models are compatible with high-throughput screening, enabling the multi-dose assessment of drug effects on both tumor viability and microvascular integrity, a critical step toward scalable therapeutic evaluation. In fact, Tung et al. constructed an NVU-GBM system to test 18 compounds and revealed differential effects on vasculature and tumor cells. While VEGF inhibitors primarily disrupted angiogenesis and CDK inhibitors showed broad cytotoxicity, standard GBM therapies, such as temozolomide and lomustine, showed limited efficacy, highlighting the need for more physiologically relevant preclinical models [58].

In conclusion, these new platforms can facilitate on-target activity assessments and will eventually be integrated with exome and single cell sequencing for comprehensive molecular profiling. As such, both 4D printed systems and NVU models hold promise for advancing precision medicine approaches in GBM by enabling scalable, rapid, and physiologically relevant drug testing to identify effective chemotherapy and chemoradiotherapy combinations. Therefore, it is easy to predict that GBM organoid models will be used for more clinical trials, hopefully guiding personalized clinical treatment decisions in the near future.

## 5. Modeling GBM in Space

### 5.1. Rationale of GBM Research in Open Space

Brain tumor organoids represent a powerful platform for basic and translational research on GBM, and their applications include GBM biology modeling, drug screening, and immunotherapy development. On the other hand, a promising field of GBM organoid applications, which is still unexplored, concerns space-based organoid brain research.

Space represents a peculiar microenvironment to which astronauts and pilots are regularly subjected. During both short- and long-term space flights, the human body is exposed to a number of stressful stimuli, including acceleration force, vibration, hypoxia, radiation, weightlessness, and microgravity [59,60].

Since space exploration has rapidly expanded in the 21st century, the issue of how space impacts human physiological and pathological processes is extremely pertinent. In fact, investigating cellular behavior in this unique environment has a significant translational potential: data obtained from experiments under microgravity or radiation could reveal information about cell functions, molecular pathways, and responses to therapy that would otherwise remain unknown on the Earth. In this context, culturing GBM organoids under spaceflight conditions, such as aboard the International Space Station (ISS), offers a unique opportunity to recapitulate in vivo-like tissue architecture and cellular dynamics with enhanced fidelity. The integration of space-based platforms into GBM research holds the potential to overcome current modeling limitations and to expedite the identification of effective, targeted therapeutic strategies for this highly aggressive malignancy.

Numerous physiological and pathological alterations in the human body result from spaceflight, including immune system weakness, muscle atrophy, loss of bone density, cardiovascular dysfunction, endocrine disorders, and space movement disorder [61]. In particular, the impact of microgravity on cancer cells has attracted the attention of the scientific community. Deng et al. exposed U251 cells to low gravity for varied lengths of time in order to examine the impact of simulated microgravity (s-μg) on glioma growth. Using a CCK8 assay to measure cell proliferation, they discovered that s-μg inhibited U251 cell activity in a time-dependent manner, meaning that the longer the cells were in microgravity, the less active they were. In particular, U251 cell death was markedly induced by 48–96 h of microgravity, which decreased cell activity to about 45% [62]. In addition, Shi et al. examined the impact of modeled microgravity (MMG) on the invasion and migration of U87 human GBM cells using a 2D-clinostat microgravity model. They found that MMG significantly reduced migration and invasion. This effect was associated with the downregulation of Orai1 expression in U87 cells and a reduction in store-operated calcium entry (SOCE) induced by thapsigargin (TG) [63]. These experiments performed on 2D models paved the way for further investigations on more sophisticated models, such as organoids.

More recently, Overbey et al. presented a comprehensive multi-omic and clinical analysis of biological responses to short duration, high elevation (585 km) spaceflight, using data from commercial and non-commercial astronaut cohorts. This study introduced the first aerospace medicine biobank (SOMA) and provided a platform for ongoing biomedical research involving private astronauts. Utilizing an array of omics technologies, including single-cell RNA and ATAC sequencing, cfRNA, spatial transcriptomics, and proteomics, this study characterized the widespread but largely transient molecular changes induced by spaceflight, with some persistent effects observed up to three months post-mission. Among the most relevant findings were significant alterations in gene expression, immune activation, DNA damage response, and oxidative stress, as well as telomere elongation and chromatin remodeling, particularly in T cells and CD14+/CD16+ monocytes. These changes mirror observations from longer missions and suggest that re-adaptation to the Earth initiates restorative molecular programs, specific to both individual cell types and shared across cell populations. Furthermore, these results support the nuanced analyses of tissue-specific shedding and transcriptional changes, and highlight the necessity of high resolution, multi-site sampling for the accurate characterization of spaceflight-related physiological shifts, especially affecting drug metabolism, immune regulation, and genomic stability. Significantly, the SOMA biobank, which includes banked biospecimens (DNA, RNA, serum, stool, etc.), is publicly accessible for expanded research and represents a critical step toward mitigating biomedical risks for future lunar, Martian, and exploration-class missions by providing a scalable framework for astronaut health monitoring and therapeutic development [64].

However, the application of similar novel methodologies to GBM research cannot be excluded.

### 5.2. Recent Advances in GBM Organoid Modeling in Space

Although GBM-specific organoid studies in space are still in their infancy, a number of studies have highlighted the incredible potential of using GBM organoids beyond the limits of the Earth to better understand the behavior of such a complex disease (Figure 4).

In 2024, Marotta et al. first conducted an important and revolutionary research on the International Space Station (ISS) in low Earth orbit (LEO) to explore the effects of microgravity on the central nervous system. They generated neural organoids from iPSCs obtained from patients with progressive multiple sclerosis or Parkinson’s disease and non-symptomatic controls, by differentiating them toward cortical and dopaminergic fates, respectively. Organoids were placed in cryovials containing a culture medium and then launched to the ISS on the SpaceX 19th Commercial Resupply Services mission for NASA (SpX CRS-19). They were cultured for approximately 1 month simultaneously on the Earth and in LEO, without a medium change. After 30 days, organoids were returned for post-flight analysis. Interestingly, organoids cultured on the ISS showed a reduced expression of genes related to cell proliferation, with increased levels of neural maturity-associated genes, compared to ground controls. In particular, cortical organoids in LEO showed low expression of cell cycle-associated genes, including *CCND2* and *CDKN2A*, as well as of early neural precursor markers (*PAX6*, *SOX2*) compared to ground cortical organoids. In addition, a protein secretion analysis revealed in LEO organoids low levels of SFRP1, a modulator of Wnt/β-catenin pathway involved in cell proliferation and migration. Similar results were found in dopaminergic organoids, with an increase of the neuronal maturity markers dopa decarboxylase (*DDC*) and tyrosine hydroxylase (*TH*) in LEO. These results demonstrated that the neural progenitor cells in these organoids differentiated faster than those on the Earth, in terms of accelerated maturation, not of accelerated aging. Moreover, organoids’ viability was similar between the two groups, while LEO-cultured organoids showed low levels of stress-related and inflammatory genes. Collectively, the results suggested that space modulates neurodevelopmental kinetics, and future ongoing experiments in this unique environment on even more complex organoids aim to better understand the actors involved in the onset of neurodegenerative diseases on the Earth, to find opportune treatments [65] (Figure 4A).

Another important study was conducted by Garcia et al., in 2023 [66], in which tumor-initiating cells obtained from GBM tissue and cultured as 3D models were launched on a suborbital, parabolic rocket flight by EXOS Aerospace Systems & Technologies. After the flight, both ground controls (GC) and rocket flight (RF)-exposed cells were analyzed and in vitro assays showed increased migratory capabilities and stemness of RF cells over controls, while the proliferation remained unchanged. In addition, RF cells, when implanted in mice, induced the growth of larger tumor-associated cystic regions and reduced host survival compared to GC cells. These data suggested that the quick exposure to extreme gravitational transitions during the suborbital flight induces a more aggressive tumor phenotype among GBM stem-like cells. The increased migratory and stemness features, as well as the enhanced tumorigenicity in vivo, demonstrated that environmental stressors, like microgravity and hypergravity, may favor the survival of more malignant clones or trigger adaptive changes in GBM stem cells [66] (Figure 4B). As stated by the authors, while considerable results concerning phenotype changes in altered gravity conditions were obtained, the investigation of the potential genetic and molecular actors involved in these changes is still lacking and represents a future area of inquiry. However, a number of pathways could be activated or enhanced during suborbital flights. GBM is characterized by a necrotic core surrounded by tumor cells under hypoxic gradients, a feature faithfully recapitulated in GBM organoids. These cells, in order to escape hypoxic conditions and to migrate to oxygen-rich areas, activate pro-migratory and pro-invasive factors, including hypoxia-inducible factor 1 (HIF-1), metalloproteinases (MMP-9, ADAM-17) [67], and chemokine receptors (CXCR4, CXCR7) [68]. The microgravity experienced during suborbital flights may simulate the hypoxic GBM niche and boost HIF-1 pathways leading to an invasive phenotype. HIF-1 is involved in angiogenesis by upregulating VEGF promoting abnormal vessels formation and, consequently, tumor aggressiveness. In addition, HIF-1 supports the expression of MMP-2/9 and CXCR4, leading to ECM remodeling and cell migration [69]. Finally, it is well established that HIF-1 signaling promotes GSCs stemness by upregulating self-renewal genes (*OCT4*, *SOX2*, *CD133*) [70], accordingly with preliminary results obtained by Garcia et al. These molecular pathways could be implicated in the enhanced tumor phenotype of cells after the suborbital flight observed in this work, and they need to be investigated in further studies.

These findings may have a great impact on GBM research conducted on the Earth. The exacerbated cancer plasticity and malignancy under peculiar conditions of gravity and hypergravity, usually missing in conventional in vitro systems, could enhance the current knowledge of GBM pathophysiology. In fact, integrating space data into terrestrial GBM models would increase their field of application, supporting the identification of unknown molecular pathways as novel therapeutic targets and a more faithful representation of the complex behavior of tumors observed in vivo.

Since one of the main responsibilities of GBM aggressiveness and resistance to therapy is the presence of TME, in March 2024, Burchett et al. developed peculiar organoids to mirror the interactions of cancer cells and immune cells in a 3D setting (Figure 4C). They created a so-called tumor–myeloid organoid, combining GBM cells and macrophages, then treated them in order to induce pro-inflammatory (M1-like) or anti-inflammatory (M2-like) phenotypes. These organoids were launched aboard SpaceX-30 to the ISS, and experiments were performed after 40 days. Although data analysis is still ongoing, the preliminary findings have revealed that microgravity helped the formation of intact, uniform, and reproducible organoids with a hypoxic core and peripherally live cells, faithfully recapitulating the TME of GBM [71].

In the context of GBM research, these pioneering models will provide significant understanding into the role of immune cells on tumor biology that would remain unexplored in Earth’s gravity, enabling the investigation of immune evasion strategies and the identification of potential immune-based therapies.

Finally, in 2023, our group participated to the Virtute-1 mission by Virgin Galactic, the first commercial suborbital spaceflight with research purposes operated by the Italian Air Force, with the aim of investigating the potential effects of a space microenvironment on GBM pathophysiology. During the suborbital flight, a number of in vitro GBM models were launched in space, including GBM spheroids and organoids. The analysis of epigenetic, transcriptomic, and metabolomic experiments is still ongoing, and the results will be the object of upcoming manuscripts.

## 6. Conclusions

In recent years, organoids have emerged as a promising tool in glioma research, due to their ability to faithfully model the features and behavior of GBM, overcoming the limitations of conventional approaches. This review provides an in-depth description of the main characteristics and protocols of GBM organoids, highlighting the pros and cons of their application in the field of basic, clinical, and translational GBM research.

However, the potential of these models is not limited to the Earth, since space-medicine offers a unique opportunity to deeper investigate the pathophysiology of complex diseases.

To the best of our knowledge, to date, no studies have yet performed a side-by-side comparison of glioblastoma organoids sent in LEO and suborbital flight, and therefore the differential effects of duration, acceleration, microgravity, and radiation remain to be elucidated. Since space-based organoid brain research is still an emerging field, further studies and in-depth investigations will be required to generate supplementary data in order to strengthen the translational potential of glioblastoma organoids.

Therefore, we have collected the latest evidence of neural and GBM organoids sent in space for the first time and, taken together, these preliminary findings lay the groundwork for advancing GBM research on the Earth by integrating space-based experimental insights into conventional cancer research, offering new perspectives on tumor biology, immune interactions, and therapeutic interventions.

## Figures and Tables

**Figure 1 ijms-26-10664-f001:**
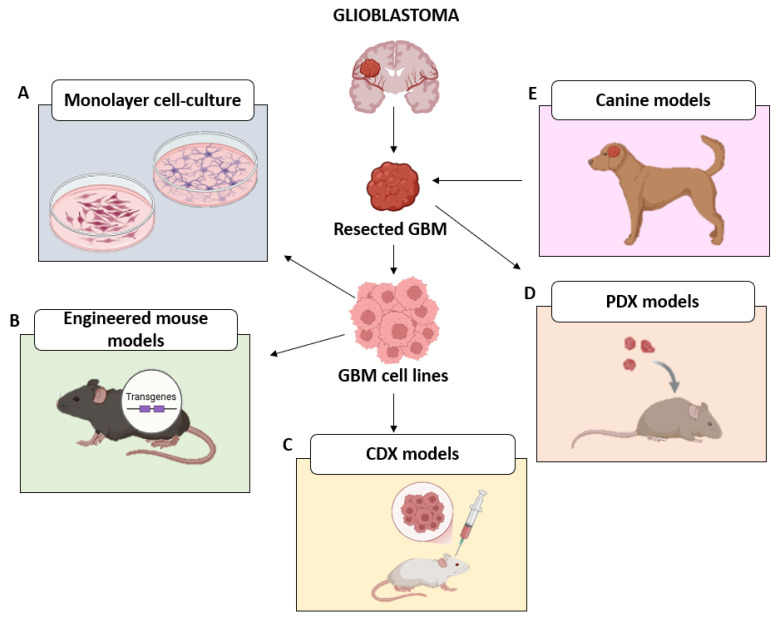
Schematic overview representing the traditional models used in GBM research. (**A**) GBM cells obtained from a patient-resected tumor cultured as a 2D monolayer adherent cell culture in a medium plus 10% serum on flat plastics: to date, the human-derived cell lines generally employed in GBM preclinical research are U87MG and U251MG. (**B**) Genetically engineered mice model (transgenic/knockout mouse) developed to investigate molecular pathways and factors involved in GBM pathogenesis. (**C**) Cell line-derived xenograft (CDX) models generated by transplanting established GBM cell lines into immunodeficient mouse. (**D**) Patient-derived xenograft (PDX) models generated by transplanting GBM fragments into immunodeficient mouse. (**E**) Canine models of spontaneous GBM similar to GBM found in humans.

**Figure 2 ijms-26-10664-f002:**
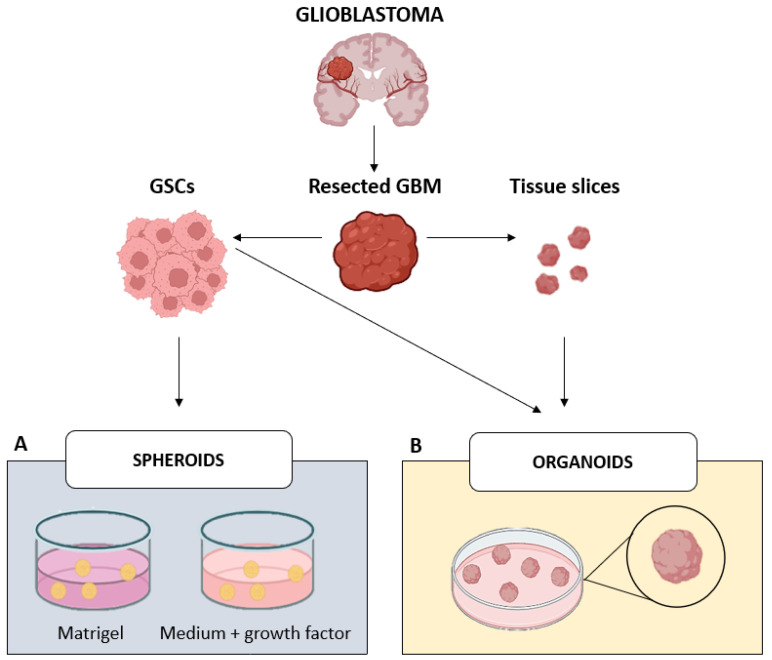
Illustration of GBM 3D in vitro models. (**A**) GSCs obtained from dissociated surgical specimens and grown as neurospheres can be cultured in specific gels, like Matrigel^®^, or in a medium supplemented with neurotrophic factors (EGF/bFGF). These models, so-called spheroids, represent the first steps of GBM research into the third dimension. (**B**) Patient-derived GSCs or tissue fragments embedded in Matrigel^®^ can be cultured in a more complex 3D-culture system, known as organoids. These models faithfully recapitulate the structure and the biology of the brain, therefore enabling an in-depth study of GBM pathophysiology and behavior.

**Figure 3 ijms-26-10664-f003:**
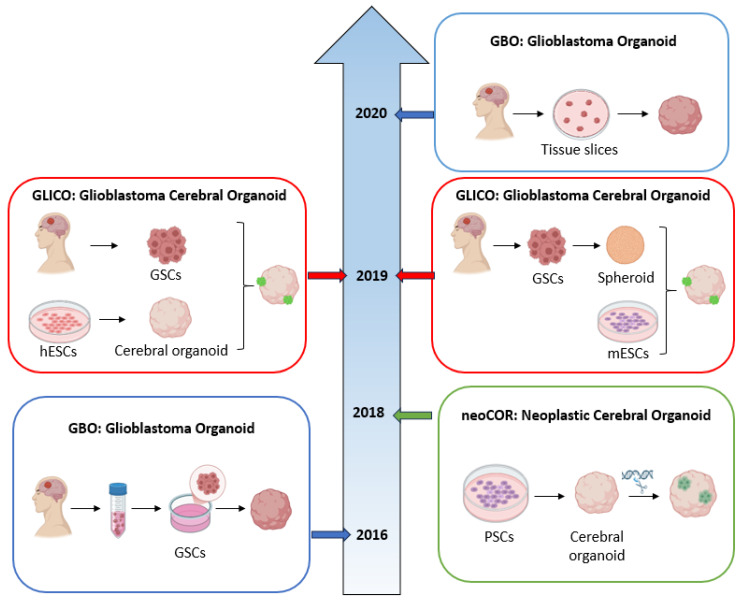
Schematic overview representing the chronological development of the main types of GBM organoids. In 2016, a GBO model generated from patient-GSCs was introduced by Hubert et al. [39], and then the protocol was reviewed and improved by Jacob et al. in 2020 [40] by replacing GSCs with patient tissue fragments. Ogawa and Bian, in 2018 [41,42], developed the neoCOR model by culturing PSCs for organoid generation followed by CRISPR-CaS9-based gene editing. GLICO models were first generated by da Silva et al., in 2018 [43], by co-culturing spheroids with mouse ESCs and, in 2019, Linkous et al. [44] set up a new protocol by co-culturing patient GSCs with cerebral organoids.

**Figure 4 ijms-26-10664-f004:**
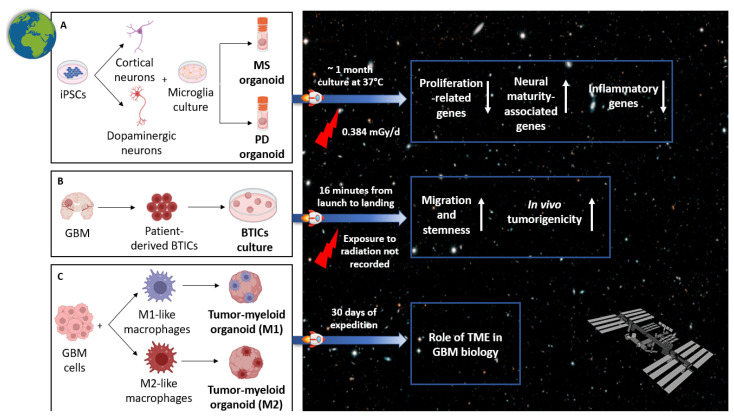
Graphical illustration of recent evidence of brain organoids launched in space. (**A**) Neural organoids developed by co-culturing iPSCs differentiated to cortical neurons (MS organoid) or dopaminergic neurons (PD organoid) with microglia and then launched in space show an increase of proliferation and neural maturity-associated genes, while low levels of stress-related genes. Organoid cultures were transported to the ISS and both ground and onboard organoids were incubated for a month at 37 °C before analysis. (**B**) GBM initiating cells isolated from patient tissue and cultured as 3D models in space show a more aggressive behavior than controls. The total time course of rocket flight was 16 min from launch to landing with approximately 39 s in hypergravity; exposure to radiation was not recorded. (**C**) Tumor–myeloid organoid generated by co-culturing GBM cells with macrophages (both M1 and M2-like) were launched aboard SpaceX-30 to the ISS. Analyses were conducted after 40 days to better understand the role of tumor microenvironment in GBM pathogenesis.

**Table 1 ijms-26-10664-t001:** Summary of advantages and limitations of different GBM models.

GBM Model	Features	Advantages	Limitations	Ref.
Traditional models				
2D model	Monolayer adherent cells	- Cost-effectiveness - Ease of use - Less ethical concerns - Reproducibility - Fast preliminary data - Suitable for high-throughput screening - Sensibility to therapy and radiation	- Lack of heterogeneity and stemness - Absence of microenvironment and interactions with non-cancer cells - Fail in mirroring native 3D architecture and spatial organization	[9,10,19]
Genetically engineered mouse model (GEMM)	Manipulated mouse model expressing GBM genetic alterations	- Enable studies of specific mutations or signaling pathways involved in tumor growth and progression	- Expensive and time-consuming model - Fail in recapitulating complexity and human microenvironment of original tumor	[22,23,45]
Cell line-derived xenografts (CDX)	Established cell line implanted into mouse brain	- Reproducibility	- Fail in recapitulating the human microenvironment of original tumor	[20]
Patient-derived xenografts (PDX)	Tissue fragments implanted into mouse brain	- Genetically stable - Allow personalized drug tests - The architecture and parental tumor features are preserved	- Expensive and time-consuming model - Fail in recapitulating the human microenvironment of original tumor	[10,19,23,45]
Canine model	Spontaneous canine GBM	- Histopathological similarities to human GBM	- Problem of detecting canine brain tumors - Ethical concerns	[10]
Innovative 3D in vitro models				
Spheroid	Glioblastoma stem cells (GSCs) cultured as neurospheres	- Native 3D architecture is preserved - Spatial organization - Allow for studies of invasiveness - Expression of stemness markers	- Lack of microenvironment and interactions between malignant and non-malignant cells - Fail in reproducing intratumoral heterogeneity - Genomic and transcriptional changes after many passages in vitro	[2,32,33]
GBO	Patient-derived GSCs cultured in Matrigel	- Both intra- and intertumoral heterogeneity are preserved - Cellular morphology, spatial distribution and hypoxic gradient are conserved	- Absence of microenvironment - No tumor–non-tumor cells interactions - No immune cells - Slow growth	[39]
	Patient-derived tissue fragments cultured in organoid medium	- Both intra- and intertumoral heterogeneity are preserved - Cellular morphology, spatial distribution, and hypoxic gradient are conserved - Original cell–cell interactions - Fast growth - Presence of immune cells and endothelial cells - Suitable model for immunotherapy tests	- Limited residual immune cells - Low vasculature	[40]
neoCOR	PSC-derived cerebral organoid genetically modified by CRISPR/Cas9 technology	- Tumor–non-tumor cells interactions - Main GBM characteristics are recapitulated - Suitable models for studies on GBM biology and drug screenings	- Lack of microenvironment - Use limited to known mutations	[41,42]
GLICO	Human GBM spheroids co-cultured with mouse ESC GSCs co-cultured with human-ESC derived cerebral organoid	- tumor–brain interactions - invasive behavior and proliferation patterns are preserved	- No immune cells - No vascularization	[43,44]

## Data Availability

No new data were created or analyzed in this study. Data sharing is not applicable to this article.

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
