# Peer review of "Modeling Glioblastoma with Brain Organoids: New Frontiers in Oncology and Space Research"

_ijms, 2025, doi:10.3390/ijms262110664_

Round 1
Reviewer 1 Report
Comments and Suggestions for Authors
This review is well written and rich in up-to-date references to the most recent literature. The paragraph dedicated to the development of preclinical models and organoids in space is particularly interesting. I believe that the work can be published in IJMS.
Author Response
Comment 1: This review is well written and rich in up-to-date references to the most recent literature. The paragraph dedicated to the development of preclinical models and organoids in space is particularly interesting. I believe that the work can be published in IJMS.
Response 1: We thank the Reviewer for the positive evaluation of our work and the recommendation for publication. We hope that the manuscript will be appreciated also in this new potentiated version.
Reviewer 2 Report
Comments and Suggestions for Authors
This review focuses on the hot topic of “GBM organoid models,” systematically integrating the three dimensions of “traditional models - 3D organoids - space research” for the first time. Notably, it combines space microgravity with GBM organoids, filling a gap in review studies within the field. It holds significant academic importance for advancing the translation of GBM basic research into clinical applications. Acceptance is recommended after revisions.
- The specific preparation protocols for the three core organoids (GBO, neoCOR, GLICO) are insufficiently described.
- Section 5.2 only reports the phenotypic effects of space microgravity on GBM cells (e.g., "enhanced invasiveness" found by Garcia 2023), but fails to deeply analyze the molecular mechanisms (e.g., whether it is related to HIF-1α or CXCL12-CXCR4 pathways) and does not compare the differential effects of different space environments (e.g., LEO vs suborbital) on GBM organoids. It is recommended to add mechanism hypotheses or literature support.
- Section 4 mentions "microfluidic chips simulating the blood-brain barrier (BBB) for drug screening", but does not provide specific case data
- Although the legends of Fig 1 (traditional models), Fig 2 (3D models), and Fig 3 (organoid evolution) are concise, the main text does not elaborate on key information for each figure (e.g., the core breakthroughs in organoid technology improvement from 2016 to 2020 in Fig 3), leading to disconnection between figures and text; Fig 4 (space research) does not label the specific treatment conditions of experimental groups (e.g., ISS culture time, microgravity intensity).
- t is recommended to unify the terminology format (full name + abbreviation for the first appearance, and unified abbreviation thereafter) to improve readability.
Author Response
Comment 1: The specific preparation protocols for the three core organoids (GBO, neoCOR, GLICO) are insufficiently described.
Response 1: We thank the Reviewer for the observation, we revised the manuscript by adding a more specific description of the protocols used to generate GBOs (lines 256-263), neoCORs (lines 291-322) and GLICOs (lines 338-343). We hope that this new revised version may fulfill IJMS’s readers.
Comment 2: Section 5.2 only reports the phenotypic effects of space microgravity on GBM cells (e.g., "enhanced invasiveness" found by Garcia 2023), but fails to deeply analyze the molecular mechanisms (e.g., whether it is related to HIF-1α or CXCL12-CXCR4 pathways) and does not compare the differential effects of different space environments (e.g., LEO vs suborbital) on GBM organoids. It is recommended to add mechanism hypotheses or literature support.
Response 2: We really thank the Reviewer for the comments. As the Reviewer has rightly pointed out, in section 5.2, particularly concerning the work of Garcia et al, no molecular pathways have been discussed: the authors focused their research on phenotypic changes of rocket-flight (RF) exposed cells compared to ground-control (GC) cells, in terms of enhanced invasiveness/aggressiveness and decreased survival in vivo. However, due to the promising preliminary results obtained, the involvement of molecular pathways deserves in-depth studies. Following reviewer’s suggestion, on the basis of existing literature we added a paragraph on potential molecular pathways and factors implicated in GBM pathogenesis, focusing on HIF-1 signaling that could be boosted by microgravity environment experienced during suborbital flights with significant effects on tumor cells behavior (Szalad et al, 2009; Armento et al, 2017; Boyd et al, 2021) (lines 626-644). Moreover, according to the reviewer’s comments we better explained gene expression results of the study conducted by Marotta et al (lines 599-605).
To the best of our knowledge, to date no studies have yet performed a side-by-side comparison of glioblastoma organoids sent in LEO and suborbital flight, and therefore the differential effects of duration, acceleration, microgravity, and radiation remain to be elucidated: since space-based organoid brain research is still an emerging field, further studies and in-depth investigations will be required to generate supplementary data (including those cited by the reviewer) in order to strengthen the translational potential of glioblastoma organoids (lines 682-687).
Comment 3: Section 4 mentions "microfluidic chips simulating the blood-brain barrier (BBB) for drug screening", but does not provide specific case data.
Response 3: We thank the Reviewer for the useful advice. We revised the manuscript by adding an example of how implementing BBB modelling in GBM organoids may be used in drug testing (Shi et al, 2023) (lines 434-445).
Comment 4: Although the legends of Fig 1 (traditional models), Fig 2 (3D models), and Fig 3 (organoid evolution) are concise, the main text does not elaborate on key information for each figure (e.g., the core breakthroughs in organoid technology improvement from 2016 to 2020 in Fig 3), leading to disconnection between figures and text; Fig 4 (space research) does not label the specific treatment conditions of experimental groups (e.g., ISS culture time, microgravity intensity).
Response 4: We really appreciate Reviewer’s comment, we implemented the explanation of Figure 3 in the main text, by adding an overview of GBM organoid development throughout the analyzed time period (lines 353-381). Moreover, as suggested, we modified Figure 4 and the relative figure legend by clarifying, where indicated, the specific information concerning treatment condition of space research models. We hope these clarifications strengthen the link between the figures and the discussion, improving both continuity and interpretability throughout the manuscript.
Comment 5: It is recommended to unify the terminology format (full name + abbreviation for the first appearance, and unified abbreviation thereafter) to improve readability.
Response 5: We thank the Reviewer for the suggestion, we have revised the manuscript so that all terms now appear with the full name followed by the abbreviation at first mention, and only the abbreviation thereafter, to enhance readability of the text.
Reviewer 3 Report
Comments and Suggestions for Authors
The present review provides a punctual, comprehensive and clear overview on an interesting scientific subject represented by glioblastoma research models. The authors properly gave a historical summary of the development of these models in glioblastoma research and their applications and provided details on the prosmising further use of GBM organoids for pre-clinical purpose and space biomedical brain reasearch. In my opinion this manuscript is well written, all the paragraphs are properly developed, and the literature is correctly cited. Overall, this manuscript represents a well done and sensible summary of the salient information in the field and therefore, in my opnion, deserves to be published on IJMS.
Author Response
Comment 1: The present review provides a punctual, comprehensive and clear overview on an interesting scientific subject represented by glioblastoma research models. The authors properly gave a historical summary of the development of these models in glioblastoma research and their applications and provided details on the promising further use of GBM organoids for pre-clinical purpose and space biomedical brain research. In my opinion this manuscript is well written, all the paragraphs are properly developed, and the literature is correctly cited. Overall, this manuscript represents a well done and sensible summary of the salient information in the field and therefore, in my opinion, deserves to be published on IJMS.
Response 1: We sincerely thank the Reviewer for her/his thoughtful and positive assessment of our manuscript. We appreciate that the reviewer recognized the completeness of our historical overview and, in particular, the relevance of our discussion of glioblastoma organoids potential in space research. We greatly appreciate the recommendation for publication in IJMS.
Reviewer 4 Report
Comments and Suggestions for Authors
In the manuscript entitled ¨Modeling Glioblastoma with Brain Organoids: New Frontiers in Oncology and Space Research¨, " the authors review in depth the diverse glioblastoma models, including the advantages and disadvantages
of passing through 2D to 4D models, including the tumor-on-a-chip approach. Also, authors describe the need to investigate the utility and projection of these models under microgravity. It is indeed required that more research in microgravity and cancer focus on the impact on human health. This review emphasizes this perspective, considering the establishment of human life in space.
The manuscript is well writed, and the figures are relevant and support it. The references are accurate. The manuscript is ready to publish.
Author Response
Comment 1: In the manuscript entitled ¨Modeling Glioblastoma with Brain Organoids: New Frontiers in Oncology and Space Research¨, " the authors review in depth the diverse glioblastoma models, including the advantages and disadvantages of passing through 2D to 4D models, including the tumor-on-a-chip approach. Also, authors describe the need to investigate the utility and projection of these models under microgravity. It is indeed required that more research in microgravity and cancer focus on the impact on human health. This review emphasizes this perspective, considering the establishment of human life in space. The manuscript is well written, and the figures are relevant and support it. The references are accurate. The manuscript is ready to publish.
Response 1: We deeply appreciate the Reviewer’s positive comments and supportive evaluation of our work. We are pleased that the reviewer found the manuscript clear, well-structured, and relevant to the field, and that the discussion on glioblastoma organoids related to space research was well received. We are pleased to receive the recommendation for publication.